# FAITHFUL AND STABLE NEURON EXPLANATIONS FOR TRUSTWORTHY MECHANISTIC INTERPRETABILITY

## ABSTRACT

Neuron identification is a popular tool in mechanistic interpretability, aiming to uncover the human-interpretable concepts represented by individual neurons in deep networks. While algorithms such as Network Dissection and CLIP-Dissect achieve great empirical success, a rigorous theoretical foundation remains absent, which is crucial to enable trustworthy and reliable explanations. In this work, we observe that neuron identification can be viewed as the *inverse process of machine learning*, which allows us to derive guarantees for neuron explanations. Based on this insight, we present the first theoretical analysis of two fundamental challenges: (1) **Faithfulness:** whether the identified concept faithfully represents the neuron's underlying function and (2) **Stability:** whether the identification results are consistent across probing datasets. We derive generalization bounds for widely used similarity metrics (e.g. accuracy, AUROC, IoU) to guarantee faithfulness, and propose a bootstrap ensemble procedure that quantifies stability along with **BE** (Bootstrap Explanation) method to generate concept prediction sets with guaranteed coverage probability. Experiments on both synthetic and real data validate our theoretical results and demonstrate the practicality of our method, providing an important step toward trustworthy neuron identification.

## 1 INTRODUCTION

Despite the rapid development and application of deep neural networks, their lack of interpretability raises growing concerns (Samek et al., 2017; Zhang et al., 2021). A popular strategy to "open the black-box" is to analyze internal representations at the level of individual neurons and associate them with human-interpretable concepts. This process is known as **neuron identification** in the field of mechanistic interpretability, which yields *neuron explanations* (Bau et al., 2017; Oikarinen & Weng, 2023). Over the past few years, many neuron identification methods have been proposed. For example, Bau et al. (2017) use curated concept datasets to identify the corresponding concept, while Oikarinen & Weng (2023) leverage multimodal models to automatically generate neuron explanations. A growing body of methods has been developed to identify concepts corresponding to neurons (Srinivas et al., 2025; Huang et al., 2023; Gurnee et al., 2023; Mu & Andreas, 2020; La Rosa et al., 2023; Zimmermann et al., 2023; Bykov et al., 2023; Kopf et al., 2024; Shaham et al., 2024).

Despite rapid empirical progress, systematic comparison and rigorous theoretical understanding of neuron identification remain limited. Recently, Oikarinen et al. (2025) unified the evaluation of neuron identification methods within a single mathematical framework to enable fair comparisons. However, deeper theoretical foundations are still lacking, which undermines the trustworthiness and reliability of neuron explanations. Consider a chest-X-ray model that predicts pneumonia and attributes its decision to a neuron purportedly representing lung opacity, when in fact the neuron responds to hospital-specific markings. Such unfaithful explanations can mislead clinicians, lead to harmful treatment decisions, and ultimately erode trust.

These concerns motivate a closer examination of the core obstacles to trustworthy neuron explanations. In particular, we identify two central challenges in current neuron identification methods:

1. **Faithfulness.** *Does the identified concept truly capture the neuron's underlying function?*

2. **Stability.** *Is the identified concept consistent across different probing datasets?*

Both challenges are closely connected with probing datasets, which are an essential component of neuron identification methods that determines the stimuli used to measure neuron activity. However, their influence is often overlooked and not rigorously examined. To address these challenges, we provide a theoretical analysis grounded in a key observation: *neuron identification can be (roughly) viewed as an inverse process of learning*. This perspective highlights structural parallels between neuron identification process and traditional machine learning, enabling us to adapt tools from statistical learning theory to formally analyze the effect of probing datasets and bound the performance of neuron identification methods.

Our contributions are summarized as follows:

**1. New insights for neuron identification.** We are the first to show that neuron identification can be viewed as an inverse process of learning, revealing structural parallels with traditional machine learning. This insight is non-trivial: it enables us to import and adapt tools from statistical learning theory to rigorously analyze key questions in neuron identification that prior work could not address, including the impact of probing datasets.

**2. Rigorous guarantees for explanation faithfulness.** We establish the first theoretical guarantees for the faithfulness of neuron explanations, answering the critical question of when a concept identified by a neuron-identification algorithm can be trusted. Our analysis is derived under a general framework, making the results applicable to most existing neuron identification methods. Simulation studies demonstrate that our theory allows quantitative analysis of how factors such as probing dataset size, concept frequency, and similarity metrics affect performance.

**3. Quantifying stability of explanations.** We present the first formal analysis of probing datasets, an essential yet previously overlooked component that determines the stimuli used to measure neuron activity. Using a bootstrap ensemble over probing datasets, we quantify the stability of neuron explanations and design a procedure to construct a set of possible concepts for each neuron, with statistical guarantees on the probability of covering the true concept.

The remainder of this paper is organized as follows: Sec. 2 formalizes the notion of neuron identification. Sec. 3 provides a rigorous analysis of the faithfulness of neuron explanations with high probability guarantees. Sec. 4 quantifies the stability of neuron identification algorithms and establishes statistical guarantees.

## 2 FORMALIZING NEURON IDENTIFICATION

In this section, we introduce the formal definition of neuron identification and the notations used in Sec. 3 and 4. Although we use the term "neuron" identification for simplicity, the framework also accommodates larger functional units within the network. Examples include a linear combination of neurons (i.e., a direction in representation space), a feature in a Sparse Autoencoder (Cunningham et al., 2023), a direction derived by TCAV (Kim et al., 2018), or a linear probe (Alain & Bengio, 2016). Below, we formally define neuron representation and concept:

**Neuron representation** $f(x) : \mathcal{X} \to \mathbb{R}$: A neuron representation is a function mapping an input $x \in \mathcal{X}$ to an activation value. Here, $\mathcal{X}$ denotes the input space (e.g. images [1]). For example, a neuron in an MLP maps the input to a scalar value. For general neural networks, the output may not be a single real number, e.g. for convolutional neural networks (CNN) $f(x)$ is a 2-D feature map. For simplicity in similarity calculation, existing works often conduct pooling (avg, max) to aggregate the feature into a single real value.

**Concept label** $c(x)$: In the literature of neuron identification (Bau et al., 2017; Oikarinen & Weng, 2023), a concept is usually defined as a human-understandable text description. For example, "cat" or "shiny blue feather". Although intuitive, this definition is not a formal mathematical definition. In this work, we define concepts as a function: a concept $c(x) : \mathcal{X} \to [0, 1]$ is a function that takes images as input, and outputs the probability of the concept. This definition is consistent with the previous works: for example, Bau et al. (2017); Bykov et al. (2024) use human annotations which output 1 if the concept is present, otherwise 0. Oikarinen & Weng (2024) use SigLIP (Zhai et al., 2023) to automatically estimate the probability that concept $c$ appears.

---

[1]The input could also be audio (Wu et al., 2024) or text (Huang et al., 2023; Gurnee et al., 2023). In this work we focus on vision models.

To search for a concept that describes the neuron representation, different methods use different measures (e.g. IoU (Bau et al., 2017), WPMI (Oikarinen & Weng, 2023), AUC (Bykov et al., 2024) and F1-score (Gurnee et al., 2023)). Interestingly, these different methods can all be described by a general similarity function $\mathsf{sim}(f, c)$, which is a functional measuring the similarity between concept $c(x)$ and neuron representation $f(x)$. With the similarity function, the neuron identification problem can be formulated as:

$$\hat{c}(x) = \arg\max_{c(x) \in C} \mathsf{sim}(f(x), c(x)) \tag{1}$$

where $C$ is the concept set (a function space under our concept definition). In our formal definition, $\mathsf{sim}(f, c)$ is a functional that takes two functions $f$ and $c$ as input, e.g. accuracy, correlation, IoU, etc. In practice, most works replace the function $f(x)$ and $c(x)$ with their realization $f(x_i)$ and $c(x_i)$ on a probing dataset $D_{\mathrm{probe}}$ as an empirical approximation, where $x_i$ is sampled i.i.d. from the underlying distribution. For example, the similarity function of accuracy is defined as the probability that two functions have the same value: $\mathsf{sim}(f, c) = \mathbb{P}(f(x) = c(x))$. When utilizing a probing dataset $D_{\mathrm{probe}}$, we can get an unbiased empirical estimation $\hat{\mathsf{sim}}(f, c; D_{\mathrm{probe}})$ for $\mathsf{sim}(f, c)$:

$$\hat{\mathsf{sim}}(f, c; D_{\mathrm{probe}}) = \frac{1}{|D_{\mathrm{probe}}|} \sum_{i=1}^{|D_{\mathrm{probe}}|} \mathbf{1}(f(x_i) = c(x_i)). \tag{2}$$

Under this approximation, the neuron identification can be formulated as the following optimization problem:

**[Neuron identification]** $\qquad \hat{c} = \arg\max_{c \in C} \quad \hat{\mathsf{sim}}(f, c; D_{\mathrm{probe}})$

$\qquad\qquad\qquad$ where $\hat{\mathsf{sim}}(f, c; D_{\mathrm{probe}}) = \hat{\mathsf{sim}}(f(x_i), c(x_i)), \ x_i \in D_{\mathrm{probe}}.$

$$\tag{3}$$

Eq. 3 shows that $D_{\mathrm{probe}}$ plays a critical role in this approximation, yet a rigorous analysis of its effect is still lacking. We address this gap in this work in Sec. 3.2 and 4.

**Why do we choose similarity-based definition?** Similarity provides a broad and unifying notion of a neuron's concept: many existing definitions can be expressed as special cases of similarity with appropriate functions. For example, a common practical criterion is that *a neuron represents concept c if its activation can successfully classify concept c.* This criterion can be formulated as a similarity function using standard classification metrics such as F1-score (Huang et al., 2023), AUC (Kopf et al., 2024), recall (Zhou et al., 2014) and accuracy (Koh et al., 2020).

## 3 THEORETICAL GUARANTEES FOR EXPLANATION FAITHFULNESS

In this section, we address a key question in neuron identification: *When can we trust a neuron explanation produced by a neuron-identification algorithm?* We begin with an important observation: **neuron identification can be viewed as an inverse process of machine learning** in Sec. 3.1. This perspective enables us to derive formal guarantees for explanation faithfulness in Sec. 3.2 and, building on these results, to quantify the stability of neuron explanations in Sec. 4.

### 3.1 ANALOGY BETWEEN NEURON IDENTIFICATION AND MACHINE LEARNING

From the formulation in Eq. 3, we observe that the neuron identification problem closely parallels supervised learning problem. Given a standard classification task and a neural network model $h \in H$, where $H$ denotes the hypothesis space containing all possible neural network models, the problem can be formalized as minimizing the loss $L$, which is typically approximated by the empirical loss $\hat{L}$ on a training dataset $D_{\mathrm{train}}$ as follows:

**[Machine learning]** $\qquad \hat{h} = \arg\min_{h \in H} \quad \hat{L}(h; D_{\mathrm{train}})$

$\qquad\qquad\qquad$ where $\hat{L}(h; D_{\mathrm{train}}) = \hat{L}(h(x_i), y(x_i)), \ x_i \in D_{\mathrm{train}},$

$$\tag{4}$$

and $y(x)$ denotes the label function and $h(x)$ is the neural network. Comparing Eq. 4 and Eq. 3, we see that these two problems share a similar structure: Both are optimization problems with objectives

Figure 1: Analogous relationship between neuron identification and machine learning. Neuron identification searches for a concept matching a neuron, while machine learning searches for a model matching human labels. Thus, neuron identification can be viewed as inverse of learning process.

of similar form. The left panel of Fig. 1 compares the procedures of these two domains, while the right panel lists their detailed correspondences. As illustrated in Fig. 1, neuron identification can be roughly viewed as the inverse process of machine learning: during learning, we search for neural network (parameters) $h(x)$ that approximates a target human concept $y(x)$ (e.g. ImageNet classes), whereas neuron identification instead searches for concept $c(x)$ (or a simple combination of concepts) that best matches a specific neuron representation $f(x)$.

Importantly, this observation enables us to leverage and adapt tools from machine learning theory while extending them to the unique setting of neuron identification. In the following, we first develop formal guarantees for the **faithfulness** of neuron explanations in Sec 3.2, and then extend this perspective to perform uncertainty quantification and assess **stability** in Sec. 4.

## 3.2 THEORETICAL GUARANTEES FOR NEURON EXPLANATIONS

In this section, we address the **faithfulness** challenge: *Does the identified concept truly capture the neuron's underlying function?* Using the framework introduced in Sec. 2, this question reduces to asking whether the identified concept truly achieves high similarity $\mathsf{sim}(f, c)$ to neuron representation. Building on the analogy between neuron identification and machine learning established in Sec. 3.1, we develop a new generalization framework tailored to the neuron identification setting. Although inspired by classical learning theory (Shalev-Shwartz & Ben-David, 2014), our analysis provides the first formal guarantees on the concept-neuron similarity $\mathsf{sim}(f, c)$. We first define the generalization gap $g$ for neuron identification as:

$$g(D_{\mathrm{probe}}, C, f) \triangleq \sup_{c \in C} [\hat{\mathsf{sim}}(f, c; D_{\mathrm{probe}}) - \mathsf{sim}(f, c)]. \tag{5}$$

We show that this gap $g(D_{\mathrm{probe}}, C, f)$ can be bounded in Thm. 3.1 under two mild assumptions: (i) the concept set $C$ is finite, and (ii) the probing dataset $D_{\mathrm{probe}}$ is sampled i.i.d. These conditions are met by most existing neuron identification methods, e.g., Bau et al. (2017); Oikarinen & Weng (2023); Bykov et al. (2024).

**Theorem 3.1.** *With probability at least $1 - \delta$,*

$$\sup_{c \in C} |\hat{\mathsf{sim}}(f, c; D_{\mathrm{probe}}) - \mathsf{sim}(f, c)| \leq r(f, D_{\mathrm{probe}}, \frac{\delta}{|C|}), \tag{6}$$

*where $r(f, D_{\mathrm{probe}}, \delta)$ describes the convergence rate of similarity function $\hat{\mathsf{sim}}(f, c; D_{\mathrm{probe}})$ and satisfies*

$$\mathbb{P}\left[\left|\hat{\mathsf{sim}}(f, c; D_{\mathrm{probe}}) - \mathsf{sim}(f, c)\right| \geq r(f, D_{\mathrm{probe}}, \delta)\right] \leq \delta. \tag{7}$$

*In Eq. 6, the confidence parameter $\delta$ is adjusted using a union bound, replacing $\delta$ with $\frac{\delta}{|C|}$.*

**Corollary 3.2.** *With probability at least $1 - \delta$,*

$$\mathsf{sim}(f, \hat{c}) \geq \mathsf{sim}(f, c^*) - 2r(f, D_{\mathrm{probe}}, \frac{\delta}{|C|}), \tag{8}$$

*where $\hat{c}$ is selected concept using Eq. 3 and $c^* = \arg\max_{c \in C}[\textsf{sim}(f, c)]$ is the optimal concept.*

**Discussion.** Thm. 3.1 adapts classical generalization theory to the neuron identification setting, where the objective of interest is $\textsf{sim}$ and $\hat{\textsf{sim}}$. This provides the first theoretical result on the $\textsf{sim}(f, c)$, which is enabled by our key insight in Sec. 3.1. The convergence rate function $r(f, D_{\text{probe}}, \delta)$ characterizes how fast the estimator $\hat{\textsf{sim}}$ converges. In Sec. 3.2.1, we will derive convergence rates for several popular similarity functions, showing that for many commonly used similarity estimators $r(f, D_{\text{probe}}, \delta) = \mathcal{O}(\sqrt{\frac{-\log \delta}{|D_{\text{probe}}|}})$. On the other hand, Corollary 3.2 suggests that by maximizing similarity on the probing dataset, the identified concept $\hat{c}$ is *approximately optimal*, within a gap determined by the convergence rate of the similarity function and the size of the concept set $C$. This result guarantees that the concept identified with the probing dataset truly achieves high similarity to the target neuron representation.

### 3.2.1 CONVERGENCE RESULTS FOR POPULAR SIMILARITY METRICS

From Thm. 3.1 and Corollary 3.2, we see that the convergence rate is a key factor controlling the generalization gap. Therefore, in this section, we derive and examine the convergence rate of common similarity metrics. Table 1 summarizes several common similarity scores and their convergence rate $r$:

1. **Accuracy:** This similarity function is used in (Koh et al., 2020), and the convergence rate of accuracy can be estimated via the Hoeffding's inequality.

2. **AUROC:** This similarity function is used in (Bykov et al., 2023), and the convergence rate is related to concept frequency $\rho(\underline{c})$ and can be derived using Thm. 2 in Agarwal et al. (2004). Fig. 3a plots the convergence rate $r_{\text{AUROC}}$ under different $\rho$ and shows that when both $\rho$ and $|D_{\text{probe}}|$ are small, the convergence rate $r_{\text{AUROC}}$ blows up, indicating imbalanced probing datasets may cause larger generalization error and reduce explanation faithfulness.

3. **Recall, precision, IoU:** These similarity functions are used in (Zhou et al., 2014), (Srinivas et al., 2025), (Bau et al., 2017) respectively. To derive their convergence rates, we view these metrics as conditional versions of accuracy: for example, precision can be regarded as computed only on examples where $f(x) = 1$. Thus, the convergence rate is similar to $r_{\text{Acc}}$, differing only in that the effective sample size changes from $|D_{\text{probe}}|$ to $(F_{11} + F_{10})$. The same reasoning applies to Recall and IoU. In practice, users can collect additional data until the effective sample size reaches desired level. Further details are provided in Appendix D.

**Summary.** So far, we have derived the generalization gap $g$ for several popular similarity metrics. These results enable practitioners to select an appropriate metric based on available probing data and the properties of the concepts. For example, our experiments in Sec. 3.3 show that AUROC converges quickly when concept frequency is high, but much slower when the frequency is low;

| sim Metric | $\textsf{sim}(f, c)$ | $\hat{\textsf{sim}}(f, c)$ | $r(f, D_{\text{probe}}, \delta)$ |
|---|---|---|---|
| Accuracy | $\mathbb{P}(f(x) = c(x))$ | $\frac{\sum_{x \in D_{\text{probe}}} \mathbf{1}(f(x)=c(x))}{|D_{\text{probe}}|}$ | $\sqrt{\frac{\log(\frac{2}{\delta})}{2|D_{\text{probe}}|}}$ |
| AUROC | $\mathbb{P}(f(x) < f(y) \mid c(x) = 0, c(y) = 1)$ | $\frac{\sum_{\{x\|c(x)=0\}} \sum_{\{y\|c(y)=1\}} \mathbf{1}[f(x)<f(y)]}{\|\{x\|c(x)=0\}\|\|\{x\|c(x)=1\}\|}$ | $\sqrt{\frac{\log(\frac{2}{\delta})}{2\rho(\underline{c})(1-\rho(\underline{c}))|D_{\text{probe}}|}}$ |
| IoU | $\frac{W_{11}}{W_{01}+W_{11}+W_{10}}$ | $\frac{F_{11}}{F_{01}+F_{11}+F_{10}}$ | $\sqrt{\frac{\log(\frac{2}{\delta})}{2(F_{11}+F_{10}+F_{01})}}$ |
| Recall | $\frac{W_{11}}{W_{01}+W_{11}}$ | $\frac{F_{11}}{F_{01}+F_{11}}$ | $\sqrt{\frac{\log(\frac{2}{\delta})}{2(F_{11}+F_{01})}}$ |
| Precision | $\frac{W_{11}}{W_{10}+W_{11}}$ | $\frac{F_{11}}{F_{10}+F_{11}}$ | $\sqrt{\frac{\log(\frac{2}{\delta})}{2(F_{11}+F_{10})}}$ |

Table 1: Similarity metrics $\textsf{sim}(f, c)$, estimation $\hat{\textsf{sim}}(f, c)$ and their corresponding convergence speed $r(f, D_{\text{probe}}, \delta)$. For simplicity, denote $W_{ij} = \mathbb{P}(f(x) = i, c(x) = j), \; i, j \in \{0, 1\}$, $F_{ij} = \frac{\{|f(x)=i,c(x)=j|x \in D_{\text{probe}}\}|}{|D_{\text{probe}}|}$. For AUROC, $\rho(\underline{c})$ is the portion of positive examples in the probing dataset $D_{\text{probe}}$ (i.e. the frequency of concept).

in such cases, switching to other similarity metric can reduce the generalization gap and improve performance.

### 3.3 SIMULATION STUDIES

To verify the theory developed in Sec. 3.2 and to compare different similarity metrics, we conduct simulations on a synthetic dataset that contains ground-truth similarity values and allows us to simulate a variety of settings. Specifically, we use binary concept $c(x) \in \{0, 1\}$ for simplicity. Neuron activations $f(x)$ are binarized by setting top-5% activations to 1 and remaining to 0. The joint distribution of $f, c$ is controlled by the probability matrix $M$: $M_{ij} = \mathbb{P}(f(x) = i, c(x) = j)$, $i, j \in \{0, 1\}$.

We conduct two experiments: (1) a **single-concept study** to compare convergence speeds and (2) a **multi-concept simulation** to verify Thm. 3.1.

**Experiment 1: Convergence speed.** In Thm. 3.1, the key factor that controls the gap is the convergence rate $r$. To investigate this, we generate synthetic data and compare different similarity functions. For the concept, we study the following two settings:

- **Setting 1:** $M =$
$$\begin{array}{cc} & \begin{array}{cc} c = 0 & c = 1 \end{array} \\ \begin{array}{c} f = 0 \\ f = 1 \end{array} & \left( \begin{array}{cc} 0.93 & 0.02 \\ 0.02 & 0.03 \end{array} \right) \end{array}$$

    This case simulates a regular concept.

- **Setting 2:** $M =$
$$\begin{array}{cc} & \begin{array}{cc} c = 0 & c = 1 \end{array} \\ \begin{array}{c} f = 0 \\ f = 1 \end{array} & \left( \begin{array}{cc} 0.9499 & 0.0001 \\ 0.0491 & 0.0009 \end{array} \right) \end{array}$$

    This simulates a rare concept (frequency is $0.001$), which often occurs when the concept is fine-grained.

We simulate with $N_{\text{exp}} = 1000$ randomly sampled datasets and plot how the 95% quantile of error changes with the number of samples, as shown in Fig. 2. From the simulation results, we can see that

1. Accuracy has the fastest convergence in both cases. On regular concept, IoU, recall and precision are similar. AUROC converges faster than them.

2. For rare concept, the convergence pattern differs: AUROC and recall are much worse than precision and IoU. This matches our analysis in Sec. 3.2, where we showed that AUROC converges much more slowly when the concept frequency is low.

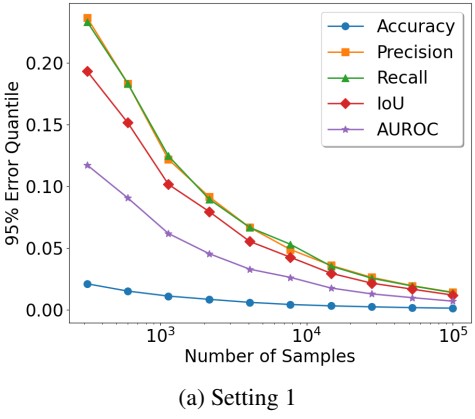
(a) Setting 1

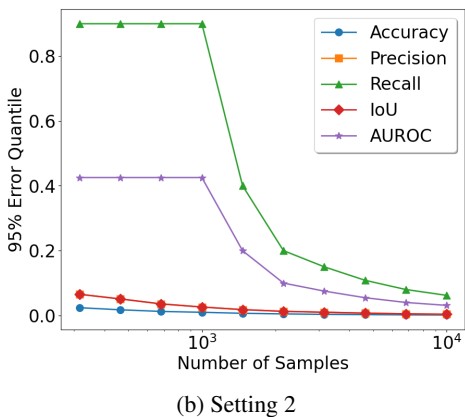
(b) Setting 2

Figure 2: 95% quantile of error of 5 similarity metrics under two settings: (a) balanced concept frequency; (b) low concept frequency (0.001). Accuracy converges fastest in both settings.

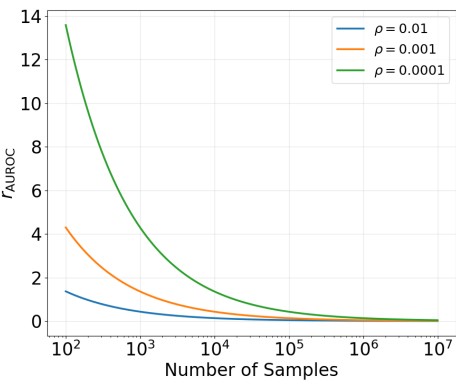
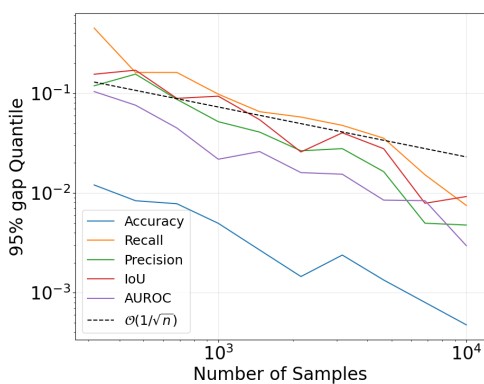

(a) Convergence rate $r_{\text{AUROC}}$ with respect to probing dataset size $|D_{\text{probe}}|$ under different concept frequency $\rho(\underline{c})$.

(b) Simulation of the generalization gap predicted by Thm. 3.1 versus probing dataset size, showing an empirical convergence rate of $\mathcal{O}(1/\sqrt{n})$.

Figure 3: Theoretical and simulation results on generalization gap.

**Experiment 2: Gap simulation**  In this experiment, we further verify Thm. 3.1 via synthetic data. Different from **Experiment 1** which simulates single concept, this test requires a concept set $C$. We generate the synthetic data with the following steps:

1. **Generate neuron representation.** Binarized neuron representation $f(x)$ is generated by setting the top-5% of activations to 1 and the rest to 0, i.e. $M_{10} + M_{11} = 0.05$.

2. **Generate concepts.** We generate $|C| = 1000$ concepts as the candidate set. For each concept $c_i$, we first generate its frequency $\mathbb{P}(c_i(x) = 1) = M_{01} + M_{11}$ from a log-uniform distribution in the interval $(10^{-4}, 10^{-1})$. Then, we sample $M_{11} = \mathbb{P}(f(x) = 1, c_i(x) = 1)$ uniformly from $(0, \min[\mathbb{P}(f(x) = 1), \mathbb{P}(c_i(x) = 1)])$ to ensure validity. Given $M_{11}$, the remaining part of $M$ can then be inferred from concept frequency and activation binarization. Given the probabilities, we compute corresponding conditional probability $(\mathbb{P}(c_i(x) \mid f(x))$ and sample $c_i(x)$ accordingly.

3. **Experiment and simulation.** We repeat the above steps $N_{\text{exp}} = 1000$ times. We use the sampled neuron representation $f(x)$ and concept activation $c_i(x)$ to calculate similarity and select top-ranked concept $\hat{c}$. Then, we compute the ground-truth similarity with the real probability matrix $M$ and calculate the error as the difference between similarity of selected concept and max similarity in the candidate set $(\max_{c \in C}[\text{sim}(f, c)] - \text{sim}(f, \hat{c}))$. We take the 95% quantile of error among all experiments to approximate the bound under success probability $1 - \delta = 95\%$.

In Fig. 3b, we plot the simulated gap against the size of the probing dataset $|D_{\text{probe}}|$. We observe that: (1) All curves have similar slope to the reference $\mathcal{O}(\sqrt{1/n})$ curve, suggesting an asymptotic convergence rate of $\mathcal{O}(\sqrt{1/n})$, which is consistent with our theoretical analysis. (2) For the constant term, accuracy has the fastest convergence and AUROC is the second. This matches our simulation of $r$ in **Experiment 1, Setting 1**, supporting our conclusion.

In summary, the simulation experiments empirically validate the correctness of our theory and show its potential to help users choose appropriate similarity metric under different settings.

## 4    QUANTIFYING STABILITY IN NEURON EXPLANATIONS

In this section, we address the second key challenge in neuron identification methods – **stability**: *Is the identified concept consistent across different probing datasets?* Leveraging the connection established in Sec. 3.1, we adopt a *bootstrap ensemble* approach for stability estimation. This method is applicable to any neuron identification algorithm without modifying its internal mechanism. Building on this bootstrapping framework, we further design a method to construct a prediction set of candidate concepts that contains the desired concept with guaranteed probability.

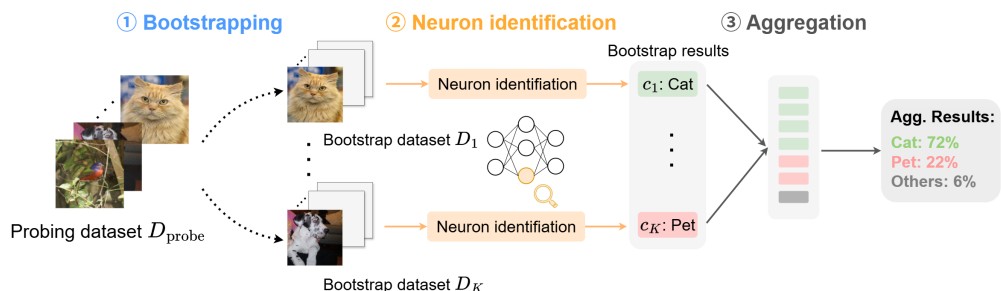

Figure 4: Illustration of bootstrap ensemble in neuron identification. Multiple probing datasets are generated via bootstrapping. Then, neuron identification algorithm is applied to each dataset and final concepts are aggregated to estimate the probability of each concept.

## 4.1 EMPIRICAL MEASUREMENT VIA BOOTSTRAP ENSEMBLE

Bootstrap ensemble (Breiman, 1996) is a machine learning technique used to improve prediction accuracy and quantify uncertainty. The method aggregates multiple models, each trained on a different resampled version of the original dataset obtained via bootstrapping (sampling with replacement). The final prediction is typically determined by majority voting, and the confidence is estimated as the proportion of models voting for the final prediction (Lakshminarayanan et al., 2017).

For neuron identification, we introduce a bootstrap-based stability framework that resamples the probing dataset to produce multiple identification outcomes for a single neuron. This adaptation allows us to quantify the stability of the neuron explanations obtained. The procedure is:

1. **Collect bootstrap datasets:** Sample $K$ datasets $\{D_i\}_{i=1}^K$ independently by randomly selecting samples from the probing dataset $D_{\text{probe}}$ with replacement.

2. **Run neuron identification:** Apply the neuron identification algorithm to each bootstrap dataset $D_i$ and record the predicted concept $c_i$.

3. **Aggregate predictions:** After $K$ runs, estimate the probability of each concept as: $\mathbb{P}(c) = \frac{1}{K}\sum_{i=1}^K \mathbf{1}(c_i = c)$, where $\mathbf{1}(\cdot)$ denotes the indicator function.

Fig. 4 summarizes the pipeline. With bootstrap ensemble, the algorithm now outputs probability of each candidate concept.

## 4.2 THEORETICAL GUARANTEES VIA CONCEPT PREDICTION-SET CONSTRUCTION

While bootstrap ensembles provide an empirical measure of stability, we also seek theoretical guarantees on the identified concept. In particular, we want to bound the probability that the most frequent concepts in bootstrap ensemble capture the desired concept[2]. To achieve this, we construct a **concept prediction set**, a set of concepts that are likely to describe the neuron, rather than a single best guess. This prediction-set approach can be applied to any neuron identification algorithm without any modifications. We call this method **Bootstrap Explanation (BE)** and list the full procedure in Alg. 1 in Appendix A.

The following theorem gives a probabilistic guarantee that a desired concept $c^*$ will be included in the prediction set constructed via the bootstrap ensemble, under mild assumptions on the candidate set and similarity function:

1. $c^* \in C$ (the desired concept is included in candidate concept set).

2. $\mathsf{sim}(f, c^*) \geq \mathsf{sim}(f, c) + \Delta, \forall c \in C, c \neq c^*$, where $\Delta > 0$ is a positive constant. This assumes the similarity function can distinguish the desired concept with other concepts.

With these assumptions, we have the following theorem:

---

[2]Analogous to the ground truth in conventional machine learning.

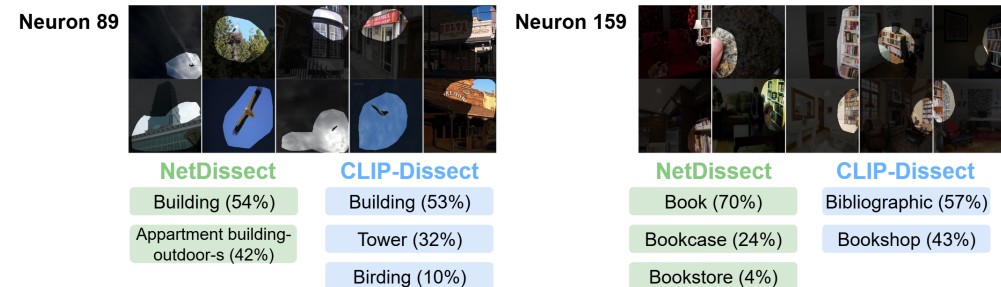

Figure 5: Results of applying bootstrap ensemble to NetDissect and CLIP-Dissect on ResNet-50 neurons. NetDissect shows more stable, concrete concepts. CLIP-Dissect outputs are more diverse and abstract. We show more results in Appendix J.1 due to space limitations.

**Theorem 4.1.** *Let $c^*$ be the desired concept for a given neuron and the assumptions above hold for $c^*$. Let $S \subseteq C$ be the prediction set constructed in Alg. 1, and let $k(S) = \sum_{i=1}^{K} [\hat{c}_i \in S]$ be the number of bootstrap trials that predict a concept in $S$. Then, under these assumptions,*

$$\mathbb{P}(c^* \in S) \geq \sum_{i=0}^{K-k(S)-1} \binom{K}{i} p^i (1-p)^{K-i}, \tag{9}$$

*where $p$ is the single-trial error probability defined implicitly by the equation $r(f, D_{\text{probe}}, \frac{p}{|C|}) = \frac{\Delta}{2}$.*

Thm. 4.1 provides a statistical guarantee on the probability that our desired concept is included in the prediction set. We postpone its proof to Appendix B.2.

### 4.3 EXPERIMENTS

We apply our BE method to two base methods: CLIP-Dissect (Oikarinen & Weng, 2023) and Net-Dissect (Bau et al., 2017). We use a ResNet-50 model trained on the ImageNet dataset (Deng et al., 2009), run $K = 100$ bootstrap samples and choose the bootstrap count threshold $t = 0.95K = 0.95 \times 100 = 95$ in Alg. 1. The results are shown in Fig. 5. In Appendix J.1, we include more results.

From the results, we can observe interesting differences between these two methods: (1) CLIP-Dissect prefers more abstract concepts. For example, it gives concepts like fostering and bibliographic. NetDissect, in contrast, tends to identify concrete concepts. (2) In general, CLIP-Dissect provides more diverse concepts and sometimes captures ones missed by NetDissect (e.g. Birding for Neuron 89). NetDissect is more stable across different bootstrap samples. A potential reason is that NetDissect utilizes localization information, which improves stability.

## 5 CONCLUSION AND LIMITATIONS

In this work, we presented a theoretical framework for neuron identification, with the goal of clarifying the **faithfulness** and **stability** of existing algorithms. Building on our key observation that **neuron identification can be viewed as the inverse process of learning**, we introduced the notion of generalization gap to quantify and derive formal guarantees for explanation faithfulness. To quantify stability, we proposed **BE** procedure to construct concept prediction sets with statistical coverage guarantees. Together, these results provide the first principled framework for the **trustworthiness** of neuron identification, complementing existing empirical studies.

Our work also has some limitations: the bound on generalization gap is a general bound for any concept set. It does not utilize the relation between concepts thus may be improved for specific concept sets. The bootstrap ensemble method provides an algorithm-agnostic way to quantify stability and generate prediction sets, but also introduces additional computational overhead.

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

# Appendix

CONTENTS

# A DETAILS ON BOOTSTRAP ENSEMBLE

---

**Algorithm 1: BE:** Generating a concept prediction set for target neuron

---

**Input:** Concept set $C$, probing dataset $D_{\text{probe}}$, target neuron $f$, neuron identification procedure `Identify`$(C, f, D_{\text{probe}})$, bootstrap sample count $K$, bootstrap count threshold $t$

**Output:** Prediction set $S$ of candidate concepts

**for** $i \leftarrow 1$ **to** $K$ **do**

 Sample dataset $D_i$ from $D_{\text{probe}}$ with replacement (same size as $D_{\text{probe}}$);

 Calculate $\hat{c}_i = $ `Identify`$(C, f, D_i)$;

**end**

For each concept $c_j \in C$, count the number of its appearances:

$$k_j = \sum_{i=1}^{K} [\hat{c}_i = c_j]$$

Sort concepts by frequency, $k_{r_1} \geq k_{r_2} \cdots \geq k_{r_s}$, $s$ is the number of different concepts generated during bootstrapping;

Initialize $S \leftarrow \emptyset$, $j \leftarrow 0$, cur_count $\leftarrow 0$;

**while** *cur_count* $< t$ **do**

 Add $c_{r_j}$ to $S$: $S \leftarrow S \cup \{c_{r_j}\}$;

 Update $j \leftarrow j + 1$, cur_count $\leftarrow$ cur_count $+ k_{r_j}$

**end**

---

## B    FORMALIZATION OF THE THEORIES

### B.1    FORMALIZATION OF THM. 3.1

**Definition 1.** *The convergence rate of similarity function is defined as a function that satisfies*

$$\mathbb{P}\left[\left|\hat{\textsf{sim}}(f, c; D_{\text{probe}}) - \textsf{sim}(f, c)\right| \geq r(f, D_{\text{probe}}, \delta)\right] \leq \delta. \tag{10}$$

*Here, $D_{\text{probe}}$ is randomly sampled from the underlying data distribution $D$ with a fixed size $|D_{\text{probe}}|$, and the probability is taken over all possible sampled probing datasets.*

*Proof of Thm. 3.1.* For each $c_i \in |C|$, from the definition of convergence rate, we have

$$\mathbb{P}\left[\left|\hat{\textsf{sim}}(f, c_i; D_{\text{probe}}) - \textsf{sim}(f, c_i)\right| \geq r(f, D_{\text{probe}}, \frac{\delta}{|C|})\right] \leq \frac{\delta}{|C|}. \tag{11}$$

Thus, with union bound, we have

$$\begin{aligned}
&\mathbb{P}\left\{\sup_{c \in C} |\hat{\textsf{sim}}(f, c; D_{\text{probe}}) - \textsf{sim}(f, c)| \leq r(f, D_{\text{probe}}, \frac{\delta}{|C|})\right\} \\
&= 1 - \mathbb{P}\left\{\cup_{c \in C} |\hat{\textsf{sim}}(f, c; D_{\text{probe}}) - \textsf{sim}(f, c)| > r(f, D_{\text{probe}}, \frac{\delta}{|C|})\right\} \\
&\geq 1 - \sum_{c \in C}\mathbb{P}\left\{|\hat{\textsf{sim}}(f, c; D_{\text{probe}}) - \textsf{sim}(f, c)| > r(f, D_{\text{probe}}, \frac{\delta}{|C|})\right\} \\
&\geq 1 - |C|\frac{\delta}{|C|} \\
&= 1 - \delta
\end{aligned} \tag{12}$$

$\square$

*Proof of Corollary 3.2.* From the definition,

$$\hat{c} = \arg\max_{c \in C} \hat{\textsf{sim}}(f, c). \tag{13}$$

Thus, $\hat{\textsf{sim}}(f, \hat{c}) \geq \hat{\textsf{sim}}(f, c^*)$. From Thm. 3.1, with probability at least $1 - \delta$, we have

$$\textsf{sim}(f, \hat{c}) \geq \hat{\textsf{sim}}(f, \hat{c}) - r(f, D_{\text{probe}}, \delta) \tag{14}$$

and

$$\textsf{sim}(f, c^*) \leq \hat{\textsf{sim}}(f, c^*) + r(f, D_{\text{probe}}, \delta). \tag{15}$$

Therefore, we have

$$\textsf{sim}(f, \hat{c}) \geq \textsf{sim}(f, c^*) - 2r(f, D_{\text{probe}}, \delta) \tag{16}$$

with probability at least $1 - \delta$. $\square$

### B.2    PROOF FOR THM. 4.1

In this section, we prove Thm. 4.1:  Theorem 4.1. *Let $c^*$ be the desired concept for a given neuron and the assumptions above hold for $c^*$. Let $S \subseteq C$ be the prediction set constructed in Alg. 1, and let $k(S) = \sum_{i=1}^{K}[\hat{c}_i \in S]$ be the number of bootstrap trials that predict a concept in $S$. Then, under these assumptions,*

$$\mathbb{P}(c^* \in S) \geq \sum_{i=0}^{K-k(S)-1}\binom{K}{i}p^i(1-p)^{K-i}, \tag{17}$$

*where $p$ is the single-trial error probability defined implicitly by the equation $r(f, D_{\text{probe}}, \frac{p}{|C|}) = \frac{\Delta}{2}$.*

*Proof.* We start the proof by estimating single-trial error rate.

**Lemma B.1.** *Let $p$ be defined implicitly by the equation*

$$r(f, D_{\text{probe}}, \frac{p}{|C|}) = \frac{\Delta}{2}, \tag{18}$$

*where $r(\cdot)$ is the uniform convergence rate in Thm. 3.1. Then,*

$$\mathbb{P}(\hat{c} = c^*) \geq 1 - p \tag{19}$$

*Remark* B.2. Lemma B.1 can be easily derived from Thm. 3.1: with probability $1 - p$, $\sup_{c \in C} |\hat{\text{sim}}(f, c; D_{\text{probe}}) - \text{sim}(f, c)| \leq \frac{\Delta}{2}$, thus

$$\hat{\text{sim}}(f, c^*; D_{\text{probe}}) \geq \text{sim}(f, c^*; D_{\text{probe}}) - \frac{\Delta}{2}$$

$$\geq \text{sim}(f, c; D_{\text{probe}}) + \frac{\Delta}{2} \quad \text{(Assumption 2)} \tag{20}$$

$$\geq \hat{\text{sim}}(f, c; D_{\text{probe}}).$$

Previously, we show that for many similarity metrics (AUROC, accuracy, IoU, etc.), $r(f, D_{\text{probe}}, \delta) = \mathcal{O}(\sqrt{\frac{-\log \delta}{|D_{\text{probe}}|}})$, i.e. $r(f, D_{\text{probe}}, \delta) \leq Q(\sqrt{\frac{-\log \delta}{|D_{\text{probe}}|}})$ for some constant $Q > 0$. In this case, we can plug in $\delta = \frac{p}{|C|}$ and get

$$\frac{\Delta}{2} = r(f, D_{\text{probe}}, \frac{p}{|C|}) \leq Q\sqrt{\frac{-\log \frac{p}{|C|}}{|D_{\text{probe}}|}}, \tag{21}$$

which gives $p \leq |C|e^{-\frac{\Delta^2}{4Q^2}|D_{\text{probe}}|}$. This shows when probing dataset size $|D_{\text{probe}}|$ and gap between desired concept and other concept $\Delta$ becomes larger, the error probability $p$ can be reduced.

Suppose we repeat our experiment $K$ times and get $\{\hat{c}_i\}_{i=1}^K$. Then, we have the following theorem.

**Theorem B.3.** *Let $k^* = \sum_{i=1}^K \mathbf{1}[\hat{c}_i = c^*]$ denotes the number of times target neuron is given during $K$ experiments. Then,*

$$\mathbb{P}(k^* \geq t) \geq \sum_{i=0}^{t} \binom{K}{i} (1-p)^i p^{K-i} \tag{22}$$

*Remark* B.4. This could be derived by Lemma B.1 and binomial distribution CDF.

Using Thm. B.3, we can derive:

$$\mathbb{P}(c^* \notin S) \leq \mathbb{P}(k^* \leq K - k(S))$$
$$= 1 - \mathbb{P}(k^* \geq K - k(S) - 1)$$
$$\leq 1 - \sum_{i=0}^{K-k(S)-1} \binom{K}{i} (1-p)^i p^{K-i} \tag{23}$$

Thus,

$$\mathbb{P}(c^* \in S) \geq \sum_{i=0}^{K-k(S)-1} \binom{K}{i} (1-p)^i p^{K-i}, \tag{24}$$

finishes the proof. $\qquad\square$

# C  RELATED WORKS

## C.1  NEURON IDENTIFICATION

The goal of neuron identification is to find a human-interpretable concept that describes the behavior and functionality of a specific neuron. A variety of methods have been proposed for neuron identification. Network Dissection (Bau et al., 2017) is a pioneering work with the idea of comparing neuron activations with ground-truth concept masks. Subsequent work explored extensions such as compositional explanations (Mu & Andreas, 2020), automated labeling with CLIP (Oikarinen & Weng, 2023), and multimodal summarization (Bai et al., 2024). More recent approaches expand the concept space to linear combinations (Oikarinen & Weng, 2024). While these advances provide useful empirical tools, in this work we aim to fill the gap in a principled theoretical foundation for neuron identification.

## C.2  PRINCIPLED FRAMEWORK FOR NEURON IDENTIFICATION

To unify the rapid growing neuron identification methods, Oikarinen et al. (2025) design a framework, summarizing most neuron identification algorithm into three major components: neuron representation, concept activations and similarity metrics. Additionally, two meta-tests are proposed to compare similarity metrics. While this work provides a good start point, rigorous theoretical analysis is still lacking, which we want to provide in this work.

## D    DETAILS IN RECALL, PRECISION AND IOU'S CONVERGENCE SPEED DERIVATION

In the main text, we mention the key idea of deriving convergence speed $r$ for recall, precision and IoU: that is regard them as special case of accuracy where data are limited in a subgroup. For recall:

$$
\begin{aligned}
\mathsf{sim}_{\text{recall}}(f, c) &= \frac{\mathbb{P}(f(x) = 1, c(x) = 1)}{\mathbb{P}(c(x) = 1)} \\
&= \mathbb{P}(f(x) = c(x) \mid c(x) = 1).
\end{aligned}
\tag{25}
$$

Therefore, we can regard calculation of recall as a rejection sampling process: The samples satisfying $c(x) = 1$ are kept and others are rejected. Then, accuracy is calculated on remaining samples. Thus, the convergence speed can be calculated by inserting the effective sample size $|\{c(x) = 1 \mid x \in D_{\text{probe}}\}|$ into the accuracy's convergence rate:

$$
r_{\text{recall}} = \sqrt{\frac{\log(\frac{2}{\delta})}{2|\{c(x) = 1 \mid x \in D_{\text{probe}}\}|}}.
\tag{26}
$$

For precision and IoU, the derivation is similar.

## E    LLM USAGE

In this article, LLM is used to check grammar and typos as well as improve the writing.

# New results for rebuttal

Below are new results we added for the rebuttal discussion:

1. **Section F:** We introduce additional details for simulation study Experiment 1 (Section 3.3).
2. **Section G:** We conduct experiments on the computational runtime and memory overhead of our Bootstrap Ensemble (BE) method.
3. **Section H:** We conduct additional experiments for our faithfulness study (Sec 3), introducing 5 more continuous metrics: AUROC, AUPRC, WPMI, MAD and correlation, with synthetic and real ImageNet validation datasets.
4. **Section I:** We study how sample number (which directly controls the generalization bound in Theorem 3.1) changes the quality of neuron explanations.
5. **Section J:** We conduct additional experiments for our Bootstrap Ensemble (BE) method, including comparison using the Broden dataset and Broden concepts, extended backbones (ViT, ConvNeXt) and quantitative study.

## F    DETAILS OF SIMULATION STUDY EXPERIMENT 1 (SEC 3.3)

**Experiment 1**    The procedure of Experiment 1 can be described as follows:

1. **Generate simulation data.** To start, we first generate the simulation data following the settings specified in Sec. 3.3. We generate paired binary variables representing ground-truth concept activations and neuron responses i.i.d. by directly sampling from the probability distribution specified by the probability matrix $M$.
2. **Calculate ground truth metrics.** Given the probability matrix $M$, the ground truth value of each metric is calculated according to their definition.
3. **Simulation.** In this step, we run simulation to simulate the convergence speed $r(\cdot)$. In each iteration $i$, we first sample a new batch of data following step 1 with size $N_{\text{sample}}$. Then, we estimate each metric using the data sample and calculate the error $err_i$. We repeat the procedure for $N_{\text{exp}} = 1000$ times, aggregate the $err_i$ in each round and report the 0.95-quantile of errors.

## G  COMPUTATIONAL COST FOR BOOTSTRAP ENSEMBLE (BE)

In this section, we study the computational cost of Bootstrap Ensemble. Theoretically, since the BE method requires $K$ times of bootstrapping and running original neuron identification algorithm, the running time should scale up linearly with $K$. In practice, however, the BE time cost is not simply $K$ times of original algorithm, as a significant portion of computation can often be shared among bootstrap samples, saving much time. Take CLIP-Dissect as an example: The CLIP-Dissect algorithm can be roughly divided into three steps:

1. Collect features from the target model and CLIP model.
2. Calculate similarity matrix between target features and CLIP features.
3. Select the final concept with highest similarity.

In the implementation of BE, the first step can be shared among bootstrap samples: we pre-compute the features of target model and CLIP on the whole probing dataset. For each bootstrap sample, we only need to fetch corresponding features, calculate similarity and select the highest similarity concept. With this, we run experiments based on a ConvNeXt-base model (Liu et al., 2022).[3] The profiling results are shown in Fig. G.1. We can see that though we do $K = 100$ bootstrap ensembles, the time overhead is only about 30% and the memory overhead is about 15%. Fig. G.2 shows how runtime changes with $K$, illustrating that our runtime is as high as $K$ times of original algorithm.
   Further, to understand the impact of concept number, we compare results on two concept sets:

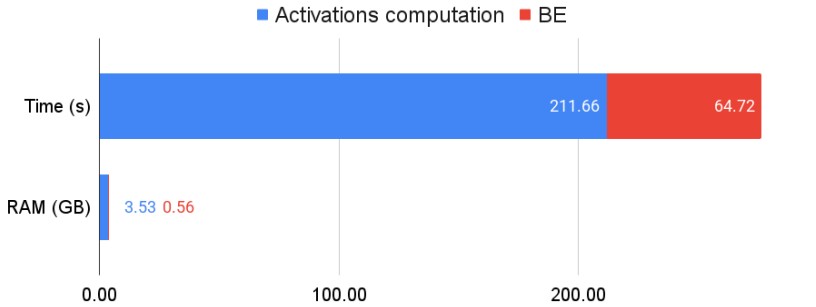

Figure G.1: Profiling result of CLIP-Dissect on a ConvNext-base model. $K = 100$ bootstrap ensembles are used.

Broden concept set with 1198 concepts and 20k English words with 20000 concepts. The result is shown in the table G.1. From the results, we can see that bootstrapping time increases significantly with larger concept set. However, we argue the major cause is the latent CLIP-Dissect uses soft-wpmi, which is much slower with larger concept set. The bootstrap wrapper itself does not introduce higher overhead.

| Stage | Broden ($|C| = 1,198$) | 20k ($|C| = 20,000$) |
|---|---|---|
| Activation computation | 211.7 | 277.6 |
| Bootstrapping | 64.7 | 348.7 |

Table G.1: Runtime (s) comparison with two concept sets: Broden and 20k English words.

---

[3]Experiment is done with an NVIDIA RTX A5000 GPU and an AMD Ryzen Threadripper PRO 3975WX CPU.

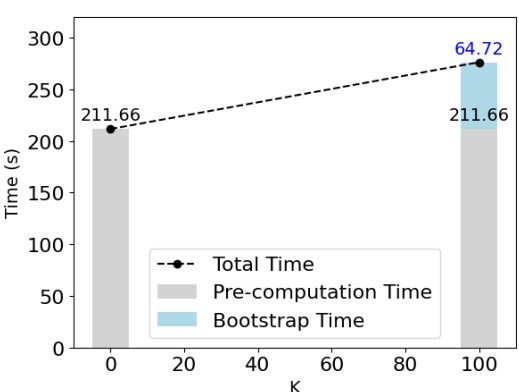

Figure G.2: Runtime vs. K for CLIP-Dissect on a ConvNeXt-base model.

## H   ADDITIONAL FAITHFULNESS EXPERIMENTS FOR CONTINUOUS METRICS

In Sec. 3.2, we mainly study bounds for metrics with binarized neuron representation. In this section, we conduct experiments on several metrics for continuous neuron representations: AUROC (Bykov et al., 2023), AUPRC, correlation (Oikarinen & Weng, 2024) , WPMI (Oikarinen & Weng, 2023) and MAD (Kopf et al., 2024). AUROC has been defined in the main text. To calculate AUPRC, we first sort $f(x_i)$ for smallest to largest: $f(x_{(1)}) \leq f(x_{(2)}) \cdots \leq f(x_{(n)})$. Thus, the precision at $k$th threshold can be defined as

$$\text{Prec}(k) = \frac{1}{k} \sum_{i=1}^{k} c_{(i)}, \tag{27}$$

The recall can be defined as

$$\text{Rec}(k) := \frac{1}{\sum_{i=1}^{n} c_i} \sum_{i=1}^{k} c_{(i)}. \tag{28}$$

The AUPRC can be calculated as:

$$\hat{\text{sim}}_{\text{AUPRC}}(f,c) := \sum_{k|c_{(k)}=1} \frac{\text{Prec}(k)}{\sum_{i=1}^{n} c_i}. \tag{29}$$

For MAD,

$$\text{sim}_{\text{MAD}}(f,c) = \mathbb{E}_{x|c(x)=1} f(x) - \mathbb{E}_{x|c(x)=0} f(x) \tag{30}$$

For WPMI, we take the definition in Oikarinen et al. (2025):

$$\text{sim}_{\text{WPMI}} = \mathbb{E}_{f(x)=1}[\log(c(x)) - \lambda \log[\mathbb{E}(c(x))]]. \tag{31}$$

Here, we take $\lambda = 1$. For correlation:

$$\text{sim}_{\text{corr}}(f,c) := \frac{\mathbb{E}\big[(f(x) - \mu_f)(c(x) - \mu_c)\big]}{\sqrt{\sigma_f^2 \sigma_c^2}}. \tag{32}$$

where $\mu_f, \mu_c, \sigma_f, \sigma_c$ are the mean and standard deviation of $f$ and $c$, respectively. For those metrics, a closed-form expression of convergence rate is challenging to derive. Thus, we conduct empirical studies with synthetic data and real ImageNet validation data.

### H.1   SYNTHETIC EXPERIMENT

**Data generation**   For the synthetic dataset, we construct a simple conditional Gaussian data-generating process to obtain pairs of binary concept activations and continuous neuron representations. For each sample $i \in \{1, \ldots, n\}$ we first randomly draw a binary concept label $c_i \in \{0, 1\}$, where $\mathbb{P}(c_i = 1) = p$ is a hyperparameter. Conditioned on $c_i$, we then sample a one-dimensional neuron representation $z_i \in \mathbb{R}$ from a Gaussian distribution whose mean and variance depend on the concept state:

$$z_i \mid c_i = \begin{cases} \mathcal{N}(\mu_{\text{pos}}, \sigma_{\text{pos}}^2), & \text{if } c_i = 1, \\ \mathcal{N}(\mu_{\text{neg}}, \sigma_{\text{neg}}^2), & \text{if } c_i = 0, \end{cases}$$

where $(\mu_{\text{pos}}, \sigma_{\text{pos}})$ and $(\mu_{\text{neg}}, \sigma_{\text{neg}})$ are hyperparameters controlling, respectively, the distribution of the neuron activation when the concept is present or absent. In practice, we generate $n$ i.i.d. samples $\{(c_i, z_i)\}_{i=1}^{n}$ according to the above process. In the experiment, we take $p = 0.002$, $\mu_{\text{pos}} = 1$, $\mu_{\text{neg}} = 0$, $\sigma_{\text{pos}} = 0.2$, $\sigma_{\text{neg}} = 0.5$.

**Experiments**   We approximate the ground truth value of metrics with $10^6$ examples, then estimate the convergence speed $r(\cdot)$ by calculating the 95% quantile of error. Since the scale of MAD is significantly different from other metrics, we compute the relative error instead. The results are shown in Fig. H.1a. From the results, we see all metrics also follow asymptotically $\mathcal{O}(1/\sqrt{N})$ convergence speed.

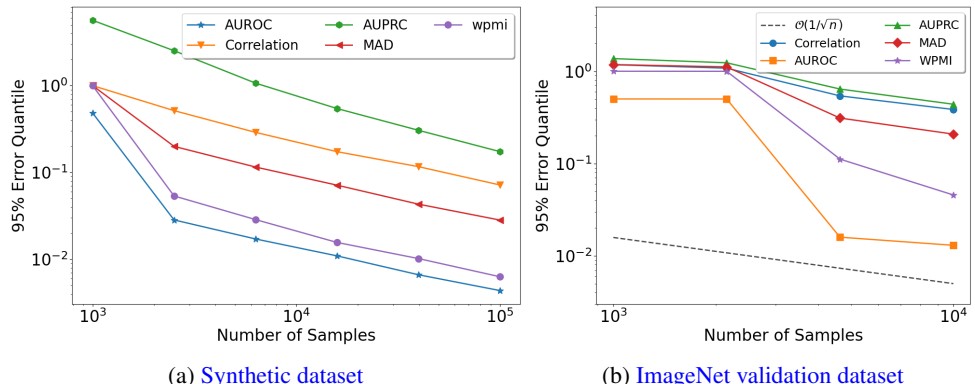

(a) Synthetic dataset

(b) ImageNet validation dataset

Figure H.1: 95% error quantile on synthetic dataset and ImageNet validation set.

## H.2 REAL DATA EXPERIMENT

Furthermore, we conduct an experiment on ImageNet validation set with a ResNet-50 model. We choose the final layer neurons and the concepts are the class labels they predict. The results are shown in Fig. H.1b. Different with the synthetic dataset, the AUROC metric performs significantly better with more samples. We hypothesize the reason is those neurons are specific to classify these concepts, thus their ability to distinguish concepts (what AUROC is measuring) are especially strong. Besides, we see that similar to synthetic data, most metrics show asymptotically $\mathcal{O}(1/\sqrt{N})$ convergence rate.

# I STUDY ON SAMPLE NUMBERS (THM 3.1)

Thm. 3.1 suggests that more probing samples can reduce the generalization gap, implying better faithfulness. To evaluate this, we conduct a qualitative study using CLIP-Dissect to study the ResNet-50 layer 4 neurons on ImageNet validation set with size of probing dataset $|D_{probe}| = 100, 1000, 10000, 50000$. We show some examples in Fig. I.1. From the examples, we can see that, with more probing data, the captured concept describes the neurons more accurately, supporting our theory.

**Neuron 2**

| N=100 | N=1000 | N=10000 | N=50000 |
|-------|--------|---------|---------|
| merchandising | slogan | signage | signage |

**Neuron 20**

| N=100 | N=1000 | N=10000 | N=50000 |
|-------|--------|---------|---------|
| sculpture | reptiles | python | python |

Figure I.1: CLIP-Dissect explanation with different number of probing samples.

# J  ADDITIONAL BOOTSTRAP ENSEMBLE (BE) EXPERIMENTS

## J.1  ADDITIONAL EXAMPLES FOR BE

In figs. J.1 to J.4, we present additional results of applying BE on NetDissect and CLIP-Dissect. To enable better comparison, we use the Broden (Bau et al., 2017) dataset as the probing dataset and corresponding concepts as concept set for both methods.

## J.2  EXPERIMENTS WITH DIFFERENT ARCHITECTURES

To understand the performance of NetDissect and CLIP-Dissect with different architectures, we add additional experiments with ConvNeXt-base (Liu et al., 2022) and ViT-B/16 (Dosovitskiy, 2020). We show example images in figs. J.5 and J.6. We compare (1) the average frequency of top-1 concept and (2) the average size of concept set that covers 90% of all runs. Since NetDissect does not support Vision Transformer features, we map the embedding back to the corresponding input patch to form a 2-D feature map. The result is shown in tables J.1 and J.2. From the results, we can see that CLIP-Dissect provides more diverse concepts except on ConvNeXt-base. NetDissect results vary a lot across backbones, which is most stable for ViT-B/16 and most diverse for ConvNeXt-base.

| Method | ResNet50 | ViT-B/16 | ConvNeXt-Base |
|---|---|---|---|
| CLIP-Dissect | 66.9% | 51.2% | 53.7% |
| NetDissect | 79.2% | 91.3% | 27.8% |

Table J.1: Comparison of top-1 concept frequency of CLIP-Dissect and NetDissect on three vision backbones.

| Method | ResNet50 | ViT-B/16 | ConvNext-Base |
|---|---|---|---|
| CLIP-Dissect | 3.11 | 5.93 | 5.61 |
| NetDissect | 2.08 | 1.34 | 6.99 |

Table J.2: Comparison of 90% coverage concept set size of CLIP-Dissect and NetDissect on three vision backbones.

**Neuron 21**

| CLIP-Dissect | NetDissect |
|---|---|
| jacuzzi-indoor-s (71%) | jacuzzi-indoor-s (95%) |
| shower stall (17%) | bathroom-s (3%) |
| baptistry-indoor-s (8%) | swimming pool (1%) |
| shower (2%) | elevator-freight_elevator-s (1%) |
| baptistry-outdoor-s (1%) | |
| ... and 1 more concepts | |

**Neuron 22**

| CLIP-Dissect | NetDissect |
|---|---|
| drinking glass (91%) | bottle (76%) |
| bubbly (8%) | bubbly (24%) |
| wineglass (1%) | |

**Neuron 23**

| CLIP-Dissect | NetDissect |
|---|---|
| planetarium-outdoor-s (78%) | sheep (61%) |
| sheep (20%) | planetarium-outdoor-s (35%) |
| geodesic_dome-outdoor-s (2%) | greenhouse-indoor-s (3%) |
| | greenhouse-outdoor-s (1%) |

Figure J.1: Additional results of applying BE to NetDissect and CLIP-Dissect on ResNet 50 neurons.

## Neuron 24

| CLIP-Dissect | NetDissect |
|---|---|
| waterfall-cascade-s (82%) | waterfall-block-s (89%) |
| waterfall (9%) | waterfall (4%) |
| fog bank (4%) | hot_spring-s (2%) |
| waterfall-block-s (4%) | waterfall-fan-s (2%) |
| steam shovel (1%) | mountain (2%) |
| | *... and 1 more concepts* |

## Neuron 25

| CLIP-Dissect | NetDissect |
|---|---|
| trouser (93%) | leg (97%) |
| studded (6%) | person (3%) |
| blue-c (1%) | |

## Neuron 26

| CLIP-Dissect | NetDissect |
|---|---|
| rocking chair (100%) | leg (62%) |
| | chair (23%) |
| | motorbike (12%) |
| | person (3%) |

Figure J.2: Additional results of applying BE to NetDissect and CLIP-Dissect on ResNet 50 neurons.

**Neuron 27**

| CLIP-Dissect | NetDissect |
|---|---|
| pleated (49%) | bird (74%) |
| curtain (28%) | movie_theater-indoor-s (15%) |
| marbled (4%) | stage-indoor-s (3%) |
| tapestry (4%) | theater-indoor_procenium-s (3%) |
| wing (4%) | person (3%) |
| *... and 4 more concepts* | *... and 1 more concepts* |

**Neuron 28**

| CLIP-Dissect | NetDissect |
|---|---|
| cat (100%) | cat (100%) |

Figure J.3: Additional results of applying BE to NetDissect and CLIP-Dissect on ResNet 50 neurons.

**Neuron 29**

| CLIP-Dissect | NetDissect |
|---|---|
| metal shutter (57%) | bookcase (82%) |
| bird cage (17%) | book (11%) |
| cage (13%) | windowpane (3%) |
| grille door (10%) | library-indoor-s (1%) |
| elevator-freight_elevator-s (1%) | reading_room-s (1%) |
| *... and 2 more concepts* | *... and 2 more concepts* |

**Neuron 30**

| CLIP-Dissect | NetDissect |
|---|---|
| spiralled (50%) | cat (72%) |
| swirly (50%) | utility_room-s (16%) |
| | laundromat-s (10%) |
| | spiralled (2%) |

Figure J.4: Additional results of applying BE to NetDissect and CLIP-Dissect on ResNet 50 neurons.

**Neuron 46**

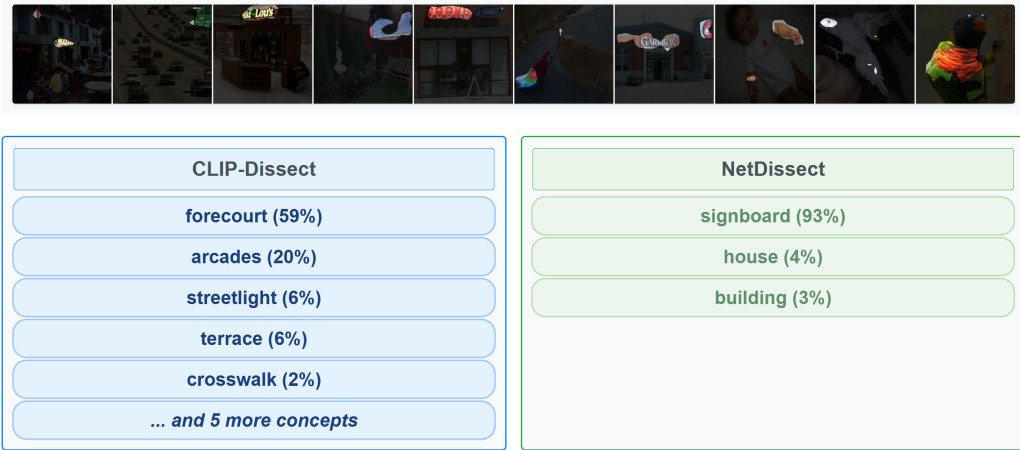

| CLIP-Dissect | NetDissect |
|---|---|
| forecourt (59%) | signboard (93%) |
| arcades (20%) | house (4%) |
| streetlight (6%) | building (3%) |
| terrace (6%) | |
| crosswalk (2%) | |
| ... and 5 more concepts | |

Figure J.5: BE example on ViT-B/16, last encoder layer.

**Neuron 70**

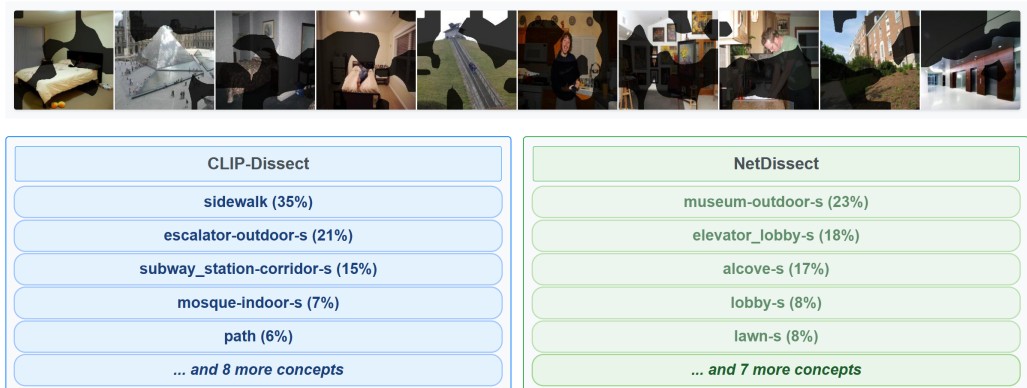

| CLIP-Dissect | NetDissect |
|---|---|
| sidewalk (35%) | museum-outdoor-s (23%) |
| escalator-outdoor-s (21%) | elevator_lobby-s (18%) |
| subway_station-corridor-s (15%) | alcove-s (17%) |
| mosque-indoor-s (7%) | lobby-s (8%) |
| path (6%) | lawn-s (8%) |
| ... and 8 more concepts | ... and 7 more concepts |

Figure J.6: BE example on ConvNext-base, last layer.

