# OpenReview forum: "Faithful and Stable Neuron Explanations for Trustworthy Mechanistic Interpretability"
_ICLR.cc/2026/Conference — Submitted to ICLR 2026_

### Official Review · Reviewer_7iEL · 2025-10-28

**Soundness:** 4
**Presentation:** 2
**Contribution:** 3
**Rating:** 4
**Confidence:** 2

**Summary:**

This paper tackles neuron identification through an inverse-learning lens: instead of training a model to fit data, it “identifies” a concept 𝑐 that best matches a fixed neuron (or neuron set) 𝑓. The authors formalize faithfulness as population-level similarity sim(𝑓,𝑐) and prove uniform generalization bounds showing that maximizing empirical similarity on a probe set yields near-optimal population similarity. They further introduce a stability notion via a bootstrap wrapper that produces prediction sets of candidate concepts with a provable coverage lower bound. The theory is instantiated for common similarity metrics (e.g., accuracy, AUROC, IoU), yielding rate expressions that translate into practical sample-size and metric-selection guidance. Empirically, they wrap standard neuron-identification methods (e.g., NetDissect/CLIP-Dissect–style pipelines) to illustrate the approach.

**Strengths:**

The authors identifie a genuine gap—neuron identification lacked a defensible definition of faithfulness—and provides a principled formulation together with a stability construct. Treating the task as an inverse problem is a fresh angle that connects interpretability with classical generalization theory. Moreover, it reframes neuron identification as an inverse problem with explicit statistical guarantees—novel for this literature.

The theory is sound and broadly applicable: uniform convergence results are presented for several practical similarity metrics, turning into actionable guidance on sample size and metric choice (e.g., dependence on concept frequency). The stability wrapper is clean, black-box, and yields interpretable prediction sets with coverage guarantees, enabling immediate integration with existing tools.

Overall, I believe the contributions of this paper exceed acceptance bar. However, due to following weakness, the authors should fix some issues to be accepted.

**Weaknesses:**

Empirical breadth and depth are insufficient.
- Experiments are concentrated on a single architecture and a narrow set of baselines; this limits external validity. Including diverse backbones (e.g., ViT/ConvNeXt, larger CNNs) and multiple concept sources would better test generality.
- The paper promises additional results in the appendix C, but I could not find no additional results at figure 6 of appendix C
- The authors should include qualitative results on how faithfulness difference affects the neuron identification quality.

Compute/efficiency analysis is missing.

- The bootstrap wrapper introduces a multiplicative factor in runtime (by 𝐾 resamples) and scales with the concept vocabulary size ∣𝐶∣. There is no systematic study of wall-clock time, memory footprint, or accuracy–efficiency trade-offs (e.g., coverage vs. 𝐾, or performance vs. ∣𝐶∣).

**Questions:**

See weaknesses above.

---

> ### Author Response · Authors · 2025-11-22
> **Response to reviewer 7iEL**
>
> Dear reviewer 7iEL,
>
> Thank you for your valuable feedback! In the rebuttal below we would like to answer your questions and address your concerns.
>
> **#1 (Weaknesses 1)** Including diverse backbones (e.g., ViT/ConvNeXt, larger CNNs) and multiple concept sources would better test generality.
>
> **A1.** Thank you for this suggestion. We have added new experiments on ViT and ConvNext in Appendix J.2 and other new experiments using CLIP with Broden concept sets in Appendix J.1. From the results, we see that CLIP-Dissect provides more diverse concepts except on ConvNeXt-base. NetDissect results vary a lot across backbones, which is most stable for ViT-B/16 and most diverse for ConvNeXt-base. This shows the generality of our method.
>
> **#2 (Weaknesses 2)** The paper promises additional results in the appendix C, but I could not find no additional results at figure 6 of appendix C
>
> **A2.** Thank you for pointing this out! We have updated the appendix and included more visualization results for multiple neurons in Appendix J.1 (Figs. J.1 - J.6).
>
> **#3 (Weaknesses 3)** The authors should include qualitative results on how faithfulness difference affects the neuron identification quality.
>
> **A3.** Thank you for the suggestion. We have added qualitative samples to show the impact of faithfulness in Appendix I: In the experiment, we vary the number of samples in probing dataset $|D_\text{probe}|$ which directly controls the faithfulness bound in Theorem 3.1. From the results, we can see that as the number of samples goes up (implies better faithfulness), the concept explanation matches the neuron better.
>
> **#4 (Weaknesses 4)** The bootstrap wrapper introduces a multiplicative factor in runtime (by 𝐾 resamples) and scales with the concept vocabulary size ∣𝐶∣. There is no systematic study of wall-clock time, memory footprint, or accuracy–efficiency trade-offs (e.g., coverage vs. 𝐾, or performance vs. ∣𝐶∣).
>
> **A4.** Thank you for bringing this up! In our three bootstrap steps (see Figure 4 in the draft), Step 1 and 3 (bootstrapping and aggregation, our bootstrap wrapper) have minimal cost. Step 2 repeats the neuron identification algorithm of interest on $K$ splits and scales with $K$. The cost of Step 2 may increase as the concept vocabulary size $|C|$ increase. Following your suggestion, we conducted a systematic study and reported the result of computation cost with different $K$ and $|C|$ in Appendix G. From the results, we see that:
> 1. The cost of bootstrapping scales linearly with $K$. However, the overall runtime is not simply $K$ times of the original algorithm. The key reason here is the costly part of computing features for the target model and CLIP can be precomputed for all data, then shared during all bootstrap samples. The overhead is only 30% for $K=100$ bootstrap samples.
> 2. For larger $|C|$, the cost of bootstrap also increases. However, we would like to clarify that this is more of a property of the neuron identification method itself (e.g. CLIP-Dissect) as bootstrapping is simply repeating the neuron identification method on each bootstrap sample. The bootstrap wrapper (Step 1 and 3) itself are much cheaper and does not scale with $|C|$.
>
> **#5 Summary**
>
> In summary, we address the reviewer’s concerns on empirical breadth and depth in A1-A3, and compute/efficiency analysis in A4:
> 1. In **A1**, we provide additional empirical results by adding ViT and ConvNext backbones in Appendix J.2 and CLIP experiments with the Broden concept set in Appendix J.1.
> 2. In **A2**, we fixed the issue and added more visualization results in Appendix J.1.
> 3. In **A3**, we add qualitative examples in Appendix I, showing that increased probing set size (reduced faithfulness bound) yields better concept explanations.
> 4. In **A4**, we clarify that runtime mainly scales with the number of bootstrap resample times $K$ and provide performance profiling in Appendix G to show our method does not introduce a significant overhead.
>
> We believe our response addresses all of the reviewer’s concerns. Please let us know if you have further questions, we would be happy to clarify and discuss further.

---

> > ### Author Response · Authors · 2025-11-26
> > **Follow-up on the rebuttal response**
> >
> > Dear Reviewer 7iEL,
> >
> > Thank you again for your valuable feedback! Since the discussion period is ending in a week, we would like to send a friendly reminder to see whether our rebuttal has addressed all your concerns.
> >
> > If you still have any questions or concerns after reading the rebuttal response, please let us know and we are willing to address them and discuss further.

---

> > > ### Comment · Reviewer_7iEL · 2025-11-26
> > >
> > > Thank you for additional quantitative results. I raised my score to 6.

---

> > > > ### Author Response · Authors · 2025-11-26
> > > > **Thank you!**
> > > >
> > > > Dear Reviewer 7iEL,
> > > >
> > > > Thank you for the positive response and raising the score to 6, we are glad that our rebuttal have addressed your concerns and questions. Thank you again for your valuable feedback to help us strengthen the manuscript.

---

### Official Review · Reviewer_Uu2E · 2025-10-30

**Soundness:** 3
**Presentation:** 2
**Contribution:** 3
**Rating:** 4
**Confidence:** 3

**Summary:**

This paper focuses on the problem of neuron identification for explanation, where the lack of formal theoretical analysis undermines the trustworthiness and reliability of existing neuron explanation methods. To overcome this limitation, the paper introduces a theoretical framework that views neuron identification as the inverse process of machine learning, enabling the adaptation of tools from statistical learning theory to establish theoretical guarantees for the faithfulness and stability of neuron explanations. The paper further conducts synthetic simulations and experiments on real-world datasets using existing neuron identification approaches to verify their theoretical findings.

**Strengths:**

1. The paper provides a theoretical analysis of neuron explanations, establishing formal bounds for various concept–neuron similarity metrics to ensure faithfulness, and deriving concept prediction probabilistic guarantees for stability.
2. The idea of treating neuron identification as an inverse machine learning process is a novel insight that enables the adaption of the generalization theory and helps justify the reliability of neuron explanations.

**Weaknesses:**

1. Although the paper presents explicit theoretical analyses, the empirical validation is limited and does not sufficiently demonstrate the effectiveness of the proposed theorems. For faithfulness, only simple binary cases and synthetic simulations are provided; for stability, the paper includes only two visualization examples, and the additional results in the Appendix appear the same to those in the main paper.
2. The paper would benefit from a more comprehensive empirical verification, including comparisons between theoretical and experimental results on real-world datasets, with quantitative and qualitative evaluations. Such analyses would substantially strengthen the support for the theoretical claims.
3. Several formal definitions of the theorems are missing, such as the convergence rate function. In addition, the simulation study details for experiments 1 and 2 are not provided in the main paper or the Appendix.

**Questions:**

1. How scalable are the similarity metrics listed in the paper? How would the proposed framework extend to other metrics, such as WPMI [1] and MAD [2]?
2. Could the paper discuss how the theoretical guarantees for neuron identification will regularize or guide the development of future neuron explanation methods?

[1] CLIP-Dissect: Automatic Description of Neuron Representations in Deep Vision Networks. ICLR 2023.

[2] CoSy: Evaluating Textual Explanations of Neurons. NeurIPS 2024.

---

> ### Author Response · Authors · 2025-11-22
> **Response to Reviewer Uu2E (1/2)**
>
> Dear reviewer Uu2E,
>
> Thank you for your valuable feedback! In the rebuttal below we would like to answer your questions and address your concerns.
>
> **#1. (Weakness 1)** …the empirical validation is limited and does not sufficiently demonstrate the effectiveness of the proposed theorems. For faithfulness, only simple binary cases and synthetic simulations are provided; for stability, the paper includes only two visualization examples, and the additional results in the Appendix appear the same to those in the main paper
>
> **A1.** Thank you for the suggestion! We would like to clarify that for faithfulness, we used binary cases because the five metrics discussed in Sec 3.2 are defined for binarized neuron representations. To address your concerns about the generalizability, we have added a new simulation study on continuous (i.e. non-binary) metrics, including AUROC, AUPRC, Correlation, WPMI, and MAD, in Appendix H.1 to demonstrate our results extend beyond the binary setting. For stability, we have followed your suggestion and added more visualization results in Appendix J.1.
>
>
> **#2. (Weakness 2)** The paper would benefit from a more comprehensive empirical verification, including comparisons between theoretical and experimental results on real-world datasets, with quantitative and qualitative evaluations. Such analyses would substantially strengthen the support for the theoretical claims.
>
> **A2.** Thank you for the suggestion! Following your suggestion, we have added additional experiments on real-world data in Appendix H.2 to further validate our theoretical results and demonstrate the broad applicability of our method. Real-data experiments are inherently more challenging because ground-truth concept annotations are rarely available. Therefore, we focus on the last-layer neurons and use the ImageNet validation set with its class labels as ground-truth concept labels. The results are shown in Appendix H.2, suggesting that AUROC performs best in this setting, while the other four metrics have similar performance. Consistent with our synthetic results, all metrics have a $\mathcal{O}(1 / \sqrt{N})$ asymptotic convergence rate.
>
> **#3. (Weakness 3)** Several formal definitions of the theorems are missing, such as the convergence rate function. In addition, the simulation study details for experiments 1 and 2 are not provided in the main paper or the Appendix.
>
> **A3.** Thank you for the feedback! Following your suggestion, we have added formal definitions and proofs for Theorem 3.1 and Corollary 3.2 in Appendix B.1 (in blue colors) and simulation study details in Appendix F.
>
> **#4. (Question 1)** How scalable are the similarity metrics listed in the paper? How would the proposed framework extend to other metrics, such as WPMI [1] and MAD [2]?
>
> **A4.** Thank you for the question! The similarity metrics we studied are scalable to large datasets (e.g. imagenet experiment in Appendix H.2). As shown in Table 1, each metric has an explicit expression and a closed-form convergence rate, making them efficient to calculate in practice.
>
> Additionally, our theoretical framework is general and readily extends to other similarity metrics such as WPMI and MAD as suggested by the reviewer. In fact, in Appendix H, we extend experiments to 5 more new metrics: MAD, AUPRC, AUROC, WPMI, and Correlation on both synthetic and real dataset. Results show that all 5 new metrics have asymptotically $\mathcal{O}(1 / \sqrt{N})$ convergence rate, which is consistent with the empirical results we get in the main text.
>
> **#5. (Question 2)** Could the paper discuss how the theoretical guarantees for neuron identification will regularize or guide the development of future neuron explanation methods?
>
> **A5.** Thank you for the suggestion! Our methods provide guidance to neuron explanation methods by:
> 1. Guiding the choice or design of new similarity function: The design of similarity function is largely heuristic in previous works [1, 4, 5]. [3] attempts to compare different similarity functions via simple sanity tests. Our work further provides quantitative criteria to measure the similarity function via its convergence, adding a new dimension to guide the selection of similarity functions. Further, we also identify potential failure cases of similarity metrics (e.g. AUROC degrades when concept is rare) from a theoretical perspective. This also helps users decide whether to use a metric based on their specific setting.
> 2. Providing new evaluation dimension: Our Bootstrap Ensemble (BE) method in Sec. 4 provides a new evaluation dimension for neuron explanation methods. This is especially important as high quality concept-labeled data is usually rare and whether the method can stably provide an explanation is important in many cases where the probing dataset is limited.

---

> > ### Author Response · Authors · 2025-11-22
> > **Response to Reviewer Uu2E (2/2)**
> >
> > **#6 Summary**
> >
> > In summary, we address the reviewer’s three concerns in A1-A3, and two questions in A4-A5:
> >
> > 1. In **A1**, we expand empirical results by adding continuous-metric simulation in Appendix H.1, real-data experiments on ImageNet validation in Appendix H.2 for faithfulness, and more visualization of our BE method in Appendix J.1.
> > 2. In **A2**, we conduct additional experiments on real data on ImageNet validation set in Appendix H.2, showing that 5 new continuous metrics also have $\mathcal{O} (1 / \sqrt{N})$ convergence rate, suggesting the wide applicability of our method.
> > 3. In **A3**, we add formal definitions and proofs for Theorem 3.1 and Corollary 3.2 in Appendix B.1, and include detailed description of simulation experiments in Appendix F.
> > 4. In **A4**, we discuss how listed metrics are scalable and extend our experiments to five additional metrics (MAD, WPMI, AUPRC, AUROC, correlation) to show the generalizability of our method.
> > 5. In **A5**, we explain how our theory guides future neuron explanation methods by (i) providing quantitative criteria for choosing or designing similarity functions and (ii) introducing BE as a new stability-oriented evaluation dimension.
> >
> > We believe our response addresses all of the reviewer’s concerns. Please let us know if you have further questions, we would be happy to clarify and discuss further.
> >
> >
> > **Reference:**
> >
> > [1] CLIP-Dissect: Automatic Description of Neuron Representations in Deep Vision Networks. ICLR 2023.
> >
> > [2] CoSy: Evaluating Textual Explanations of Neurons. NeurIPS 2024.
> >
> > [3] Evaluating Neuron Explanations: a unified framework with sanity checks. ICML 2025
> >
> > [4] Network dissection: Quantifying interpretability of deep visual representations. CVPR 2017
> >
> > [5] Natural language descriptions of deep visual features. ICLR 2021.

---

> > > ### Author Response · Authors · 2025-11-26
> > > **Follow-up on the rebuttal response**
> > >
> > > Dear Reviewer Uu2E,
> > >
> > > Thank you again for your valuable feedback! Since the discussion period is ending in a week, we would like to send a friendly reminder to see whether our rebuttal has addressed all your concerns.
> > >
> > > If you still have any questions or concerns after reading the rebuttal response, please let us know and we are willing to address them and discuss further.

---

### Official Review · Reviewer_iGbW · 2025-10-31

**Soundness:** 3
**Presentation:** 3
**Contribution:** 3
**Rating:** 4
**Confidence:** 3

**Summary:**

Recent advances, such as Network Dissection and CLIP Dissection, have achieved remarkable progress in neuron identification. However, these approaches still lack a theoretical foundation for ensuring faithful and stable explanations. The authors observe that neuron identification is similar to the inverse process of machine learning. Building on this insight, authors derive theoretical guarantees for neuron-level explanations and present theoretical analyses addressing the challenges of (1)Faithfulness and (2)Stability in neuron interpretation.

**Strengths:**

- This work presents a novel and insightful perspective by interpreting the neuron identification problem as an inverse process of machine learning. This conceptual shift provides a fresh way to reason about how neurons emerge and can be systematically explained within learned models.
- This work strengthens the theoretical foundation of neuron interpretation by addressing both faithfulness and stability. It provides analytical support for the reliability of neuron-level explanations and offers theoretical clarity beyond empirical observation.

**Weaknesses:**

- This work concludes that CLIP Dissect tends to identify more abstract concepts, whereas NetDissect captures more concrete ones. However, it remains unclear whether the probing set and concept set used for this comparison are consistent across both methods. Specifically, CLIP Dissect employs images from the model’s training data as the probing set and utilizes a concept set of approximately 2K words, while NetDissect uses the Broden dataset, which includes segmentation mask annotations but contains far fewer samples. Therefore, when applying the proposed Bootstrap Ensemble procedure under the original settings of each method, the resulting bootstrap datasets and outcomes are unlikely to be derived from identical conditions. A clarification on this point would strengthen the validity of the comparison.
- The paper introduces five similarity metrics, accuracy, AUROC, IoU, recall, and precision, as general evaluation measures. However, the applicability of each metric highly depends on the label type of the probing set and concept set. For example, IoU typically requires pixel-level ground-truth segmentation masks, making it suitable for NetDissect (which uses the Broden dataset) but not directly applicable to CLIP-Dissect, which lacks such annotations. Therefore, it would be beneficial to clarify whether different metrics are selectively used depending on the label types of the probing and concept sets if all five metrics are claimed to be generally applicable, to provide additional explanation on how each metric is adapted to different concept identification methods.

**Questions:**

- How broad is the applicability of the approach (e.g., beyond vision, beyond single neurons, beyond simple concept sets)?
- The focus is on individual neurons and concept alignment. But neurons may be polysemantic. Or representations may be distributed. How does that affect the validity of using single-neuron explanations?
- The derivation of generalisation bounds for metrics such as IoU and AUROC is an interesting contribution. Are the bounds tight/meaningful in practice?
- What are the implications of the proposed method for downstream interpretability applications (e.g., model auditing, bias detection, editing networks)?
- How many neurons/explanations were evaluated in the empirical section?

---

> ### Author Response · Authors · 2025-11-22
> **Response to Reviewer iGbW (1/3)**
>
> Dear reviewer iGbW,
>
> Thank you for your valuable feedback! In the rebuttal below we would like to answer your questions and address your concerns.
>
> **#1 (Weakness 1).** However, it remains unclear whether the probing set and concept set used for this comparison are consistent across both methods…when applying the proposed Bootstrap Ensemble procedure under the original settings of each method, the resulting bootstrap datasets and outcomes are unlikely to be derived from identical conditions.
>
>
> **A1:** Thank you for pointing this out. In Section 4, our primary goal is to assess the stability of each method **individually** with the Bootstrap Ensemble procedure, so using the same probing dataset and concept set is not strictly required for our core stability analysis. However, we agree that aligning the probing dataset and concept sets enables a fairer cross-method comparison. In response, we have added new experiments that evaluate CLIP-Dissect with the same probing data and concept set that NetDissect used (Broden dataset), making the comparison with NetDissect fully consistent. These new results are now included in Appendix J of the updated manuscript, along with additional experiments on two more backbones (ViT and ConvNext). We also report the top-1 concept accuracy and concept set size to cover 90% of runs, providing a more detailed and controlled comparison between the two methods.
>
> **#2 (Weakness 2).** It would be beneficial to clarify whether different metrics are selectively used depending on the label types of the probing and concept sets if all five metrics are claimed to be generally applicable
>
> **A2:** Thank you for the comment. To clarify, whether a metric is applied at the instance level or pixel level is not a property of the metric itself, but rather a choice in the study design. For example, although IoU is commonly used with pixel-wise labels [1][2], it can also be computed with instance-wise labels [7]; conversely, metrics such as Recall or Precision can be evaluated at the pixel level when pixel-wise concept annotations are available.
>
> In our work, the 5 metrics discussed in Sec 3.2.1 (Accuracy, AUROC, Recall, precision, IoU) are generally applicable whenever concept labels are available – whether instance-wise or pixel-wise. Our focus is on the behavior of the metrics, not on a particular label format. Importantly, our theoretical result (Theorem 3.1) is general and not restricted to these 5 metrics. We analyzed these 5 metrics in Sec 3.2.1 because their convergence rates admit clean closed-form expressions, but Theorem 3.1 applies to a broader family of metrics [7] (e.g. AUROC, correlation, etc).
>
> In Appendix H of the updated manuscript, we additionally included new results for 5 more metrics (AUROC, AUPRC, MAD WPMI and correlation) on both synthetic and real imagenet validation dataset. These results demonstrate that our method is applicable to a broader range of metrics.
>
>
> **#3. (Question 1)** How broad is the applicability of the approach (e.g., beyond vision, beyond single neurons, beyond simple concept sets)?
>
> **A3:** Our method is broadly applicable to different domains and settings, as it only requires the activation value for the unit being studied and concept labels that can be compared to the activation via any similarity metric. Below we illustrate the broad applicability of our approach in each item:
> * **Beyond vision:** Our method does not limit the input to be images. For example, the work [5] on language model representations fits naturally into our framework: neuron activations can be compared with concept labels using metrics such as F1 score, and our main theorem 3.1 applies directly.
> * **Beyond single neurons:** Our method can be used on a broader family of units (e.g. individual neurons, group of neurons, linear probing, steering vectors, concept activation vectors, sparse autoencoder features). For example, the work [6] proposed to use a Concept Activation Vector $v$ to represent a concept c for a representation $r$, which fits naturally into our framework by regarding the activation as representation’s projection on $v$: $r^T v / \|v\|_2$, with corresponding concept label c. Thus, we can apply our main theorem 3.1.
> * **Beyond the simple concept set:** Our theorem 3.1 only requires that the concept set to be finite, which is a standard setting in most practical applications. The framework does not make assumptions about the complexity, structure, or semantics of the concept set.
>
> We will include the above discussion into the updated manuscript to improve the clarity. Thank you for the question.

---

> > ### Author Response · Authors · 2025-11-22
> > **Response to Reviewer iGbW (2/3)**
> >
> > **#4. (Question 2)** But neurons may be polysemantic. Or representations may be distributed. How does that affect the validity of using single-neuron explanations?
> >
> > **A4:** Our method is applicable to both polysemantic and monosemantic neurons. When a neuron is polysemantic, our method in Sec 3 still provides a valid lower bound on the similarity score – this bound may naturally be low, reflecting the genuine ambiguity of the neuron’s behavior rather than a limitation of the method.
> >
> > Additionally, the Bootstrap Ensemble (BE) procedure in Sec 4 can help to identify poly-semantic neurons: For example, in Fig. 5 of the manuscript, Neuron 89 activates on both buildings and birds, which is captured during bootstrapping. We will incorporate this clarification into the updated manuscript to improve the clarity. Thank you for the question.
> >
> >
> > **#5. (Question 3)** Are the bounds tight/meaningful in practice?
> >
> > **A5:** Thank you for the question. Yes, the bounds are meaningful and we highlight three concrete ways below where they provide useful guidance in neuron-identification workflows:
> >
> > 1. Our bound provides a new angle to select similarity metrics: The bound suggests that when using the same probing dataset size and same concept set, we should pursue metrics that converge faster, or having smaller $r(\cdot)$. From the synthetic experiment in Sec. 3.3, we can see that the real generalization gap in Fig. 3 roughly follows the order of $r(\cdot)$ in Fig. 2. This means users can use the convergence rate $r(\cdot)$ as a metric to select a similarity function.
> > 2. Our bound provides a confidence interval for the final score: After selecting a concept, users can now not only have a final similarity score (e.g. 0.5), but also get a confidence interval from our bound (e.g. $[0.4, 0.6]$). This provides useful information about how well this concept matches the neuron.
> > 3. As shown in Table 1, our bounds suggest that all listed metrics have a generalization gap upper bound that decreases with the square root of probing dataset size. This provides useful information about how the gap changes with probing dataset size, guiding users to choose a proper size of probing dataset for their application.
> >
> >
> > **#6. (Question 4)** What are the implications of the proposed method for downstream interpretability applications (e.g., model auditing, bias detection, editing networks)
> >
> > **A6:** Thank you for asking! Our method provides a useful foundation for downstream interpretability applications by ensuring the neuron identification results are trustworthy. If the identified neuron concept is unstable or incorrect, any downstream use of that explanation may be unreliable. For example, in model auditing and bias detection applications, an unreliable neuron explanation may cause practitioners to focus on the wrong feature, leading to incorrect conclusions about model behavior or source of bias. Similarly, in model editing, an unreliable neuron explanation may cause the editor to modify the wrong feature and unintentionally change the model’s behavior.
> >
> > **#7. (Question 5)** How many neurons/explanations were evaluated in the empirical section?
> >
> > **A7:** We evaluate our method on 2048 neurons in the ResNet-50 models. Due to space limitations, we only show some examples. More examples can be found in Appendix J. Further, we add experiments on ViT-B/16 and ConvNeXt-base in Appendix J which have 768 and 1024 neurons, respectively.

---

> ### Author Response · Authors · 2025-11-22
> **Response to Reviewer iGbW (3/3)**
>
> **#8 Summary**
>
> In summary, we address the reviewer’s two concerns in A1 and A2, and five questions in A3-A7:
> 1. In **A1**, we clarify the bootstrap ensemble evaluation protocol is for individual method and add new experiments comparing CLIP-Dissect and Net-Dissect on the same dataset to enable a fairer comparison
> 2. In **A2**, we clarify that the five metrics listed in the draft are generally applicable to both instance-level and pixel-level concept labels, and show that our main theorem also applies to another 5 metrics like AUROC, MAD, AUPRC, WPMI, and correlation.
> 3. In **A3**, we demonstrate that our framework is broadly applicable beyond vision, beyond single neuron and to any finite concept set.
> 4. In **A4**, we clarify that our method works for both polysemantic and monosemantic neurons, and we show that the BE method also helps to reveal polysemantic neurons.
> 5. In **A5**, we highlight that our bounds are useful for selecting similarity metrics, providing confidence intervals, and guiding the choice of a proper probing dataset size
> 6. In **A6**, we illustrate that our method provides a reliable foundation for downstream interpretability applications by ensuring trustworthy explanations.
> 7. In **A7**, we clarify that we evaluate all the neurons and show more examples in the appendix.
>
> We believe our response addresses all of your concerns. Please let us know if you have further questions, we would be happy to clarify and discuss further.
>
> **Reference**
>
> [1] Bau et. al. Network dissection: Quantifying interpretability of deep visual representations. CVPR 2017
>
> [2] La Rosa et. al. Towards a fuller understanding of neurons with clustered compositional explanations. NeurIPS 2024
>
> [3] Koh et. al. Concept bottleneck models. ICML 2020
>
> [4] Bykov et. al. Labeling Neural Representations with Inverse Recognition. NeurIPS 2023
>
> [5] Gurnee et. al. Finding Neurons in a Haystack: Case Studies with Sparse Probing. TMLR 2023
>
> [6] Kim et. al. Interpretability Beyond Feature Attribution: Quantitative Testing with Concept Activation Vectors (TCAV). ICML 2018
>
> [7] Oikarinen et. al. Evaluating Neuron Explanations: a unified framework with sanity checks. ICML 2025

---

> > ### Author Response · Authors · 2025-11-26
> > **Follow-up on the rebuttal response**
> >
> > Dear Reviewer iGbW,
> >
> > Thank you again for your valuable feedback! Since the discussion period is ending in a week, we would like to send a friendly reminder to see whether our rebuttal has addressed all your concerns.
> >
> > If you still have any questions or concerns after reading the rebuttal response, please let us know and we are willing to address them and discuss further.

---

### Author Response · Authors · 2025-11-22
**Global response: New results and updated draft**

Dear reviewers,

Thank you all for the valuable feedback! Below is a summary of new results and clarifications during the rebuttal:
1. **Expanded architectures, datasets and more results:** In Appendix J, we provide additional Bootstrap Ensemble (BE) examples which use Broden dataset and concept set for fair comparison. Additionally, we conduct experiments with ConvNext and ViT architecture and add quantitative comparison of CLIP-Dissect vs. NetDissect by measuring top-1 frequency and concept set with 90% coverage of runs.
2. **Faithfulness experiments with more metrics and real dataset:** We extend the study to 5 additional metrics (AUROC, AUPRC, MAD, WPMI and correlation) on both a synthetic dataset and real ResNet-50 dataset in Appendix H.
3. **Qualitative study on the impact of faithfulness:** We provide qualitative results in Appendix I showing how faithfulness metric impacts the quality of explanation.
4. **Enriched details:** We add more details including formalized version and proof of Theorem 3.1. in Appendix B, and details of simulation study in Appendix F.

We highlighted all the revision and new results in blue color in the updated manuscript. The individual responses to each reviewer’s concerns and questions are posted directly under each reviewer’s review comment.

---

### Meta-Review · Area_Chair_aphc · 2026-01-07

**Summary:**

This paper proposes a principled theoretical framework for neuron identification by framing it as an inverse machine learning problem, introducing formal notions of faithfulness and stability with accompanying guarantees. Reviewers consistently found the core idea novel and theoretically well motivated. The rebuttal substantially improves the submission by adding formal definitions, broader metric coverage, additional qualitative analyses, efficiency profiling, and experiments on more architectures. Nevertheless, despite these improvements, the empirical evidence remains limited in scope and does not yet convincingly demonstrate the broad practical impact at the level expected for acceptance.

**Reviewer Concerns:**

The rebuttal partially addresses the main concerns raised by reviewers:

Additional experiments on multiple backbones and concept sources were added.

Missing formal definitions and simulation details were clarified.

Qualitative examples and a basic runtime analysis of the bootstrap procedure were included.

However, important concerns remain:

The proposed Bootstrap Ensemble (BE) method is evaluated primarily as a wrapper around two existing neuron-identification pipelines (CLIP-Dissect and NetDissect). While this is reasonable for a method-agnostic framework, the empirical study remains narrow, with the main paper focusing on a single backbone (ResNet-50 on ImageNet) and broader validation largely deferred to the appendix.

As a result, the experiments demonstrate feasibility but stop short of clearly establishing generality or strong practical impact across diverse interpretability settings.

Only one reviewer explicitly updated their score after the rebuttal, leaving uncertainty about whether the remaining reviewers would consider their concerns fully resolved.

**Reviewer Scores:**

Reviewer 7iEL: Explicitly raised the overall score from 4 to 6 after reading the rebuttal and additional experiments.

Reviewer iGbW: Did not provide a post-rebuttal update. While the rebuttal addresses their main concerns regarding metric applicability and comparison fairness, the remaining uncertainty around empirical breadth suggests the score would likely remain around 4–5.

Reviewer Uu2E: Did not provide a post-rebuttal update. Although the rebuttal improves formal clarity and adds empirical results, the reviewer’s original concerns about limited empirical validation may only be partially resolved. The score would likely remain around 4–5.

---

### Decision · Program_Chairs · 2026-01-26

Reject